# Changes in Diversity and Composition of Rhizosphere Bacterial and Fungal Community between Resistant and Susceptible Pakchoi under *Plasmodiophora brassicae*

**DOI:** 10.3390/ijms242316779

**Published:** 2023-11-26

**Authors:** Dan-Dan Xi, Lu Gao, Li-Ming Miao, Li-Ao Ge, Ding-Yu Zhang, Zhao-Hui Zhang, Xiao-Feng Li, Yu-Ying Zhu, Hai-Bin Shen, Hong-Fang Zhu

**Affiliations:** 1Shanghai Key Laboratory of Protected Horticultural Technology, Horticultural Research Institute, Zhuanghang Comprehensive Experiment Station, Shanghai Academy of Agricultural Sciences, Shanghai 201403, China; ddxi@saas.sh.cn (D.-D.X.); gaolu@saas.sh.cn (L.G.); 11616010@zju.edu.cn (L.-M.M.); 13648320910@163.com (D.-Y.Z.); szyzzh@163.com (Z.-H.Z.); lixiaofeng@saas.sh.cn (X.-F.L.); yy5@saas.sh.cn (Y.-Y.Z.); shb8311@163.com (H.-B.S.); 2Jinshan Agricultural Technology Extension Center, Shanghai 201599, China; geliaogao@163.com

**Keywords:** *P. brassicae*, fungi, bacteria, rhizosphere, diversity

## Abstract

*Plasmodiophora brassicae* (*P. brassicae*) is a soil-born pathogen worldwide and can infect most cruciferous plants, which causes great yield decline and economic losses. It is not well known how microbial diversity and community composition change during *P. brassicae* infecting plant roots. Here, we employed a resistant and a susceptible pakchoi cultivar with and without inoculation with *P. brassicae* to analyze bacterial and fungal diversity using 16S rRNA V3-V4 and ITS_V1 regions, respectively. 16S rRNA V3-V4 and ITS_V1 regions were amplified and sequenced separately. Results revealed that both fungal and bacterial diversity increased, and composition was changed in the rhizosphere soil of the susceptible pakchoi compared with the resistant cultivar. In the four groups of R_mock, S_mock, R_10d, and S_10d, the most relatively abundant bacterium and fungus was *Proteobacteria,* accounting for 61.92%, 58.17%, 48.64%, and 50.00%, respectively, and *Ascomycota*, accounting for 75.11%, 63.69%, 72.10%, and 90.31%, respectively. A total of 9488 and 11,914 bacteria were observed uniquely in the rhizosphere soil of resistant and susceptible pakchoi, respectively, while only 80 and 103 fungi were observed uniquely in the correlated soil. LefSe analysis showed that 107 and 49 differentially abundant taxa were observed in bacteria and fungi. Overall, we concluded that different pakchoi cultivars affect microbial diversity and community composition, and microorganisms prefer to gather around the rhizosphere of susceptible pakchoi. These findings provide a new insight into plant–microorganism interactions.

## 1. Introduction

*Plasmodiophora brassicae* (*P. brassicae*), an obligate biotrophic protist, causes serious cruciferous clubroot, which is spread worldwide to decrease crop yield. *Plasmodiophora,* together with *Spongospora*, *Polymyxa*, and *Sorosphaera,* constitutes *Plasmodiophorida,* which further belongs to *Phytomyxea* [1]. The pathogen can survive in soil for over 20 years in the form of resting spores [2]. At present, certain progress in the life cycle of *P. brassicae* has been made to describe the infection process [2]. Generally, the infection of *P. brassicae* would be divided into two phases: (I) the primary infection, which is restricted to the root hairs of the host, and (II) the secondary infection, which targets the cortex of the host [2]. During primary infection, pathogens release oval-shaped or pyriform biflagellate motile spores in soil with water, known as primary zoospores. These zoospores swim to and infect root hairs, forming primary plasmodia, which then cleave into zoosporangia. Later, the secondary zoospores are formed in zoosporangia and released back into the soil. The secondary zoospores penetrate the cortical tissues of the host and develop secondary plasmodia, finally causing gall formation [2]. The primary infection usually do not cause macroscopic symptoms and significant yield reduction, while the secondary infection leads to recognizable symptoms in roots [2]. The resting spores will be released into the soil after the root disintegrates to complete the pathogen lifecycle.

Physiological specialization has been well known to occur in *P. brassicae* through pathogen abilities infecting different hosts. Several differential sets, like Williams’ differential set and the European Clubroot Differential (ECD) set, have been established to identify the pathotypes of *P. brassicae* [3,4]. Furthermore, studies confirm that sequencing polymorphisms exist between high proportions of genes of different pathotypes [5]. Recently, 12 pathotypes of *P. brassicae* in Korea were identified using 22 commercial cultivars from Korea, China, and Japan, and 15 inbred lines [4]. Additionally, a specific gene was identified in pathotype 4, which may serve as potential molecular markers to recognize pathotype 4 from other pathotypes [6]. According to the ECD set, 42 pathotypes were observed in Central Europe and Sweden [7].

To date, over 3700 species of Brassicaceae are potential hosts of *P. brassicae*, including the commonly cultivated Brassica plants—*B. oleracea* L., *B. rapa* L., and *B. napus* L., which seriously restrict crop growth and yield [3]. Various *P. brassicae*-resistant crop cultivars were bred and resistant genes or loci were identified, including *Rcr1* from *B. rapa* ssp. *chinensis*, *Rc6* from *B. nigra*, *qBrCR38-1* from *B. rapa* ssp. *chinensis*, *qCRc7-4* from *B. oleracea*, and *RCA8.1* from *B. rapa* L. [8,9,10,11,12]. Despite breeding resistant crops, biological control has become an economic and ecologically friendly approach to managing clubroot. Recently, bacteria and fungi containing *Bacillus velezensis*, *Bacillus amyloliquefaciens*, *Bacillus cereus*, *Streptomyces melanosporofaciens stran X216*, *Trichoderma* Hz36 and HK37, and *Trichoderma harzianum* were found to suppress clubroot disease [13,14,15,16,17].

The rhizosphere, the area surrounding growing plant roots, is of great importance for plant health and development since it is the place where plant roots and microorganisms interact [18]. Many studies have investigated that rhizosphere microbial diversities and communities are shaped by plant genotypes and soil characteristics. For example, wild rice displayed different rhizosphere bacterial populations compared with rice cultivars [19]. Recent studies demonstrated that overall rhizosphere microbial communities significantly differed between resistant and susceptible cotton cultivars under the *Verticillium dahlia* infection [20,21]. Microbiome structures also differed between the two tomato varieties with different resistance to the soil-born pathogen *Ralstonia solanacearum*. Moreover, wild ginseng had a different diversity and structure of the rhizosphere microbial communities compared to the cultivated ginseng under cultivation modes [22].

Though many studies have demonstrated that the microbial diversities and communities in plant rhizosphere soil are associated with plant genotypes, the knowledge of the microbial diversity of different crop cultivars infected with *P. brassicae* is very limited. Even though, it has been revealed that the endosphere microbial population and relative abundance are much higher in asymptomatic than in symptomatic roots using *Brassica napus* [23]. Letreton et al. (2019) analyzed the microbial communities of roots from healthy plants and inoculated plants using a susceptible *Brassica rapa* [24]. The influence of pakchoi cultivars on rhizosphere microbial diversity infected with *P. brassicae* still remains unclear. Pakchoi (*Brassica campestris* ssp. *chinensis* Makino), one of the non-heading Chinese cabbages belonging to the Cruciferae family, is an important green leafy vegetable cultivated in China and East Asia and has become increasingly popular around the world in recent years [25]. Unfortunately, pakchoi is a host of *P. brassicae,* and most cultivars are susceptible to the pathogen. Until 2017, affected areas had reached 2500 hm^2^ in thirty-nine counties and nine towns in Shanghai [10]. Therefore, it is critical to explore the rhizosphere-associated bacterial and fungal communities to demonstrate the potential interaction between pakchoi roots and microorganisms.

The objectives of this work were (i) to analyze the diversities and communities of rhizosphere bacteria and fungi of two different pakchoi cultivars to *P. brassicae,* (ii) determine the effects of pakchoi cultivars on the composition of bacteria and fungi, and (iii) observe potential biocontrol agents. Our study firstly reported the microbial diversity in the rhizosphere of two pakchoi cultivars and found that a total of 40,150 bacterial ASVs and 390 fungal ASVs were detected from the rhizosphere of two cultivars, most of which belonged to *Proteobacteria* and *Ascomycota*. Additionally, *Mortierella* was distinctly increased in the resistant cultivar in rhizosphere soil inoculated with *P. brassicae.* This was the first time *Mortierella* was correlated with *P. brassicae,* which indicated that *Mortierella* might be a new biocontrol agent for controlling clubroot.

## 2. Results

### 2.1. Phenotypes of Pakchoi Infected with P. brassicae

To study the influences of different pakchoi cultivars on microorganism diversity in rhizosphere soil under *P. brassicae*, we employed two pakchoi cultivars, CR100 and Suzhouqing (SZQ). It was observed that both the incidence of CR100 and SZQ was 0% without inoculation (Figure 1A and Table 1). When CR100 and SZQ were inoculated with *P. brassicae* for 30 days, the incidence of CR100 and SZQ was 0% and 100%, respectively (Figure 1B and Table 1). These data showed that CR100 is a resistant cultivar, while SZQ is was a susceptible pakchoi cultivar.

### 2.2. Microbial Diversity Analysis of Rhizosphere Soil

To investigate the effect of *P. brassicae* on different pakchoi cultivars, microbial diversity analysis was performed with rhizosphere soil. The total number of reads per sample was 2,772,741 for bacteria and 2,710,321 for fungi. After filtering and deleting reads with low quality, there were 1,688,730 for bacteria and 2,161,224 for fungi. In total, the average number of genera was 237.8, 247.2, 347.6, and 363.6 obtained in R_mock, S_mock, R_10d, and S_10d for bacteria, respectively (Figure 2A), and 13.6, 14.2, 27, and 30.8 R_mock, S_mock, R_10d, and S_10d for fungi (Figure 2B). Without inoculation, the top three most relatively abundant bacteria were *Proteobacteria* (61.92%), *Gemmatimonadetes* (10.32%), and *Acidobaceria* (8.44%) in R_mock and *Proteobacteria* (58.17%), *Gemmatimonadetes* (13.22%), and *Acidobaceria* (9.39%) in S_mock (Figure 2C). The most relatively abundant fungi were *Ascomycota* (75.11% in R_mock, 63.69% in S_mock) and *Basidiomycota* (24.84% in R_mock, 36.23% in S_mock) (Figure 2D). In the samples inoculated with *P. brassicae*, the top three most abundant bacteria were *Proteobacteria*, *Acidobacteria*, and *Actinobacteria*, accounting for 48.64%, 17.28%, and 15.01% of the proportion in R_10d and 50.00%, 14.17%, and 12.83% in S_10d (Figure 2C). The top three most abundant fungi were *Ascomycota* (72.10%), *Mortierellomycota* (17.96%), and *Mucoromycota* (5.60%) in R_10d, while the top three fungi in S_10d were Ascomycota (90.31%), *Basidiomycota* (4.94%), and *Mortiereomycota* (2.78%) (Figure 2D). A change in microorganisms in proportion revealed that the ability to recruit the microorganisms of different pakchoi cultivars may be different.

### 2.3. Microbial Diversity Analysis of Rhizosphere Soil

Alpha diversity of bacteria and fungi among the four groups was estimated using the Chao, Observed_species, Simpson, Shannon, Faith_pd, Pielou_e, and Goods_coverage indexes. Without inoculation, all the indices except Goods_coverage from S_mock were similar to those from R_mock, both in bacteria and fungi, and then were significantly increased in groups inoculated with *P. brassicae* (Figure 3A,C). The Goods_coverage indices decreased in R_10d and S_10d compared with R_mock and S_mock for bacteria, and there was no significant difference between uninoculated and inoculated groups for fungi (Figure 3A,C). The most abundant microorganisms were *Dyella* for bacteria and *Trichoderma* and *Byssochlamys* for fungi both in R_mock and S_mock (Figure 3B,D). However, in inoculated groups, the most abundant bacteria and fungi turned into *Cellulomonas* and *Byssochlamys*, respectively (Figure 3B,D). These data revealed that inoculation with *P. brassicae* improved the abundance in soil and, susceptible cultivars might be profitable for microorganism growth and improve community structure around the rhizosphere.

A principal coordinates analysis (PCoA) plot showed that rhizosphere samples of R_mock and S_mock were indistinguishable, while samples of R_10d and S_10d were significantly separated for bacteria (Figure 4A). For fungi, R_mock and S_mock samples were separated, while R_10d and S_10d samples were indistinguishable (Figure 4B). However, the PCoA plot distinguished the soil samples inoculated with *P. brassicae* from uninoculated soil samples both for bacteria and fungi.

### 2.4. Species Differences Analysis

To further understand the microbial community, the ASV among different groups was calculated. In total, we obtained 40,150 bacterial ASVs and 390 fungal ASVs (Figure 5). S_mock and R_mock shared 2648 and 39 bacterial and fungal ASVs, respectively, while R_10d and S_10d shared 4154 and 82 ASVs, respectively (Figure 5A,B). The unique bacterial ASVs in R_mock, S_mock, R_10d, and S_10d were 6409, 5845, 9488, and 11,914, respectively, and fungal ASVs were 43, 50, 80, and 103, respectively (Figure 5A,B). Here, it was noticed that the number of both bacteria and fungi ASVs increased significantly in the inoculation groups compared to the uninoculated groups. Moreover, between the two inoculated groups, ASVs in the S_10d group were much higher than those in R_10d, indicating that microorganisms might be more likely to cluster around the susceptible cultivar rhizosphere.

To further identify the notable different taxa, the LefSe analysis of rhizosphere bacterial communities displayed that there were 107 differentially abundant taxa in the samples of R_mock, S_mock, R_10d, and S_10d, with values of 19, 9, 25, and 54, respectively (Figure 6A). The most abundant taxa in R_mock was *Rhodanobacteraceae*, in S_mock R_10d it was *Gemmatimonadetes*, in R_10d it was *Betaproteobacteriales*, and in S_10d it was *Bacteroidetes* (Appendix A). The LefSe analysis of fungal communities showed that there were 39 differentially abundant taxa in the samples of R_mock, S_mock, R_10d, and S_10d, with values of 4, 1, 11, and 33, respectively (Figure 6B). The most abundant taxa in R_mock was *Hypocreaceae*, in S_mock it was *Agaricomycetes*, in R_10d it was *Mortierellomycota*, and in S_10d it was *Eurotiomycetes* (Appendix A).

## 3. Discussion

Since *P. brassicae* was first identified in 1878 by Woronin, various studies were made on *P. brassicae* and its hosts [26]. However, few studies have investigated the relationship between the microbial diversity of rhizosphere soil and pakchoi cultivars in response to *P. brassicae*. We noticed that the microbial community and number were much richer in rhizosphere soil with *P. brassicae* than that without *P. brassicae* (Figure 1 and Figure 2). This phenomenon may be possibly caused by diseased roots, which carry a large of microorganisms. Some studies showed that host plants determine the bacterial and fungal community composition, like tomato, sugarcane, and cotton [21,27,28]. In our study, both fungal and bacterial ASVs in S_10d were much higher than those in R_10d (Figure 4). These data demonstrated that different pakchoi cultivars can alter microbial community and diversity in the rhizosphere under the same cultivation, supporting previous studies [21,27]. Many studies reported that plant root exudates could affect the microbial community of the rhizosphere. Four wetland plants were found to have different concentrations of root exudates that were correlated with rhizosphere microbial density and diversity [29]. Eisenhauer et al. (2017) found that root exudates promoted fungal biomass [30]. Zhang et al. (2020) found that root exudates from a resistant cultivar inhibited pathogen activity, while the exudates from a susceptible cultivar positively regulated colony growth using tobacco [31]. Untargeted metabolomics revealed that the chemical class of phenolics was abundant in the tolerant potato cultivars compared to the susceptible cultivars [32]. By extension, we speculated that the CR100 cultivar may produce different root exudate composition or concentration compared with SZQ, which further affects rhizosphere microbial diversity.

It was worth noting that *Dyella* abundance greatly decreased in the soil inoculated with *P. brassicae* (Figure 3B). To our knowledge, this is the first time that *P. brassicae* was found to inhibit *Dyella* growth. Boer et al. (2005) considered that bacteria and fungi competed for plant-derived substrates, and fungi had a strong impact on bacteria evolution [33]. This result implied that *P. brassicae* may inhibit *Dyella* growth by an unclear mechanism, which requires further study.

Furthermore, PCoA represented the differences in rhizosphere microbial communities among the S_mock, R_mock, S_10d, and R_10d. We noticed that S_mock and R_mock samples were highly similar, while S_10d and R_10d could be distinguishable for bacteria (Figure 4A). In contrast, S_10d and R_10d shared more similarities for fungi (Figure 4B). It seems like bacteria are more susceptible to be affected than fungi in the rhizosphere soil of different pakchoi cultivars with *P. brassicae*. Lebreton et al. (2019) demonstrated that *P. brassicae* strongly modified the endophytic bacterial community and the fungal community lightly [24]. It can be inferred that *P. brassicae* in the rhizosphere may interact with bacteria directly or indirectly rather than fungi, and this might be one of the mechanisms of the distinguishable difference in PcoA between S_10d and R_10d for bacteria.

Regarding the fungi, *Ascomycota* was the most relatively abundant and *Basidiomycota* was the second most relatively abundant phyla in uninoculated soil (Figure 2D), agreeing with the observations that most of the terrestrial fungi belonging to *Ascomycota* and *Basidiomycota*, and *Ascomycete* were predominant in agricultural soils [34,35]. This result was consistent with the previous observations that *Ascomycota* was the most relatively abundant fungus in the rhizosphere soil of *Brassica rapa* [24]. *Trichoderma*, belonging to *Ascomycota*, was reported as a biocontrol agent against clubroot. Two *Trichoderma*, Hz36 and Hk37, were identified to inhibit the germination of resting spores and promote root growth in rapeseed [16]. *Trichoderma harzianum* LTR-2 drastically reduced the incidence of clubroot disease in Chinese cabbage and the relative abundance of *Delftia* and *Pseudomonas* but increased other bacteria in rhizosphere soil [13]. Interestingly, *Trichoderma* and *Byssochlamys* were the main identified populations in the soil without *P. brassicae,* while *Byssochlamys* was the main identified population in the soil with *P. brassicae* (Figure 3D). *Trichoderma* was entirely undetectable in R_10d, while a small amount *Trichoderma* was detected in S_10d rhizosphere soil (Figure 3D). We presume that the zoospores of *P. brassicae* dominate in inoculated rhizosphere soil and then compete with *Trichoderma*, resulting in the inability of *Trichoderma* to survive and reproduce.

Additionally, *Mortierella* was defined as an important indicator genera of plant agronomic traits and rhizomicrobiota and was detected in the healthy roots of tumourous stem mustard, *Brassica napus*, and tomato [23,36,37]. A *Mortierella alpine* strain, ITA1-CCMA 952, isolated from *Schistidium antarctici*, produces antibiotics, antioxidants, and polyunsaturated fatty acids [38]. These data confirm that *Mortierella* is beneficial for plant health and also indicate that the higher relative abundance of *Mortierella* in R_10d rhizosphere soil may contribute to pakchoi against *P. brassicae* infection. Despite *Trichoderma*, *Mortierella* can be considered as a new potential biocontrol agent for Brassica plants in response to *P. brassicae*, which deserves to be studied.

## 4. Materials and Methods

### 4.1. Pakchoi Materials

The two inbred lines, Suzhouqing (SZQ, a susceptible pakchoi) and CR100 (a resistant cultivar), were bred in a disease nursery. After homozygous SZQ and CR100 were obtained, seeds were sown in a plastic tray (52 cm × 26.5 cm × 4 cm) with 105 pots (15 × 7 pots), and each pot (3.3 cm × 3.3 cm × 4 cm) was cultivated 1 plant. Each pot was filled with soil purchased from DAYI AGRITECH CORPORATION LIMITED (Taiwan, China), mainly containing peat, coconut husk, and lime. The trays were kept in a growth cabinet at 22 ± 2 °C, 16 h light/8 h dark, and 60–70% humidity for ten days for microbial diversity sequencing and thirty days for the observation of symptoms.

### 4.2. P. brassicae Inoculation

We used an inoculation method based on a previous study [39]. Briefly, diseased roots were collected and stored at −20 °C. The roots were washed with water and ground to extract resting spores. Then, the concentration of spore suspension was adjusted to 2 × 10^8^ and mixed with nutrient soil (DAYI AGRITECH CORPORATION LIMITED, Taiwan, China) (1000 mL suspension/1 kg soil). Pakchoi seeds were sown in the soil with or without *P. brassicae* suspension. The disease incidence was calculated after 30 days after sowing. Ten days later, rhizosphere soil was collected for microbial diversity analysis with five biological replicates of each group.

### 4.3. Microbial Diversity Sequencing

Rhizosphere soil was collected as previously reported [21]. According to the report, roots were shaken vigorously and then transferred into a 50 mL centrifuge tube that contained 15 mL 1× phosphate buffer solution. The centrifuge tube was rotated for 5 min and then the roots were removed. Tubes were centrifuged at 4000× *g* and 4 °C for 10 min, and the supernatant was discarded. Then, tubes were centrifuged at 8000× *g* for 5 min, and the supernatant was discarded. The remaining part was considered rhizosphere soil. Each biological replicate contained at least 5 g of rhizosphere soil.

Total microbial DNA from rhizosphere soil was extracted with an OMEGA Soil DNA Kit (M5635-02) (Omega Bio-Tek, Norcross, GA, USA) according to the manufacturer’s instructions. NanoDrop NC2000 (Thermo Fisher Scientific, Waltham, MA, USA) was used for detecting the quantity and quality of genomic DNA.

For bacteria, 16S rRNA genes in the V3-V4 region were amplified using the primers F: 5′-ACTCCTACGGGAGGCAGCA-3′ and R: 5′-GGACTACHVGGGTWTCTAAT-3′. PCR components contained 5 μL buffer, 2 μL dNTPs, 1 μL of each forward and reverse primer, 1 μL DNA template, and 0.25 μL Fast pfu DNA polymerase. The PCR procedure was as follows: 98 °C for 5 min, 25 cycles consisting of denaturation at 98 °C for 30 s, annealing at 53 °C for 30 s, extension at 72 °C for 45 s, and a final extension of 5 min at 72 °C. For fungi, the ITS gene V1 region was amplified with primers F: 5′-GGAAGTAAAAGTCGTAACAAGG-3′ and R: 5′-GCTGCGTTCTTCATCGATGC-3′. PCR components contained 5 μL reaction buffer, 5 μL GC buffer, 2 μL dNTPs, 1 μL of each forward and reverse primer, 2 μL DNA template, 8.75 μL ddH_2_O, and 0.25 μL Q5 DNA polymerase. The PCR procedure was as follows: initial denaturation at 98 °C for 5 min, 25 cycles consisting of denaturation at 98 °C for 15 s, annealing at 55 °C for 30 s, extension at 72 °C for 30 s, and a final extension of 5 min at 72 °C. The high-throughput sequencing of 16S rRNA and ITS amplicon was performed by Shanghai Personalbio Technology Co., Ltd. (Shanghai, China) on a Novaseq6000 platform.

### 4.4. Biodiversity Analysis

DADA2 was used for filtering, dereplication, chimera identification, and merging paired-end reads [40]. After that, amplicon sequence variants (ASVs) were obtained. Before evaluating ASV richness and alpha diversity index, rarefaction curves of 16S and ITS were presented (Appendix A). The alpha diversity of the bacterial and fungal communities was evaluated with Chao richness, Observed_species, Shannon index, Simpson index, Pielou_e index, and Good_coverage. Principal coordinate analysis (PCoA) was used based on the Bray–Curtis dissimilarity matrix to reflect beta diversity. Linear discriminant analysis (LDA) effect sizes (LefSe) were performed to detect the remarkably different taxa among samples. The LDA threshold is higher than 2.82 in bacteria and higher than 2 in fungi.

## 5. Conclusions

We conclude that pakchoi cultivars with different resistance to *P. brassicae* influence microbial diversity and composition in rhizosphere soil. In the resistant cultivar, both bacterial and fungal numbers of taxa and alpha diversity were decreased. Moreover, *Mortierellomycota* was promoted with higher abundance, while *Ascomycota* and *Basidiomycota* were inhibited. A potential biocontrol agent, *Mortierella*, was markedly increased. These results provide a better understanding of rhizosphere microorganisms in response to *P. brassicae*, which will contribute to the biocontrol of clubroot.

## Figures and Tables

**Figure 1 ijms-24-16779-f001:**
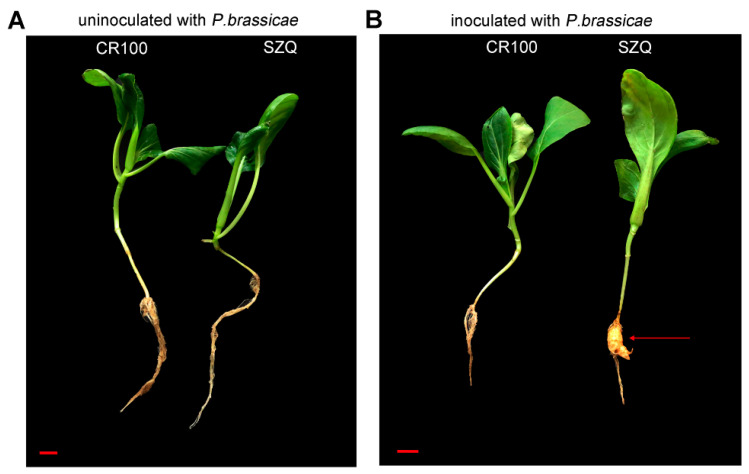
Phenotypes of pakchoi without (**A**) and with (**B**) *P. brassicae* inoculation for 30 days. SZQ represents Suzhouqing, a susceptible pakchoi; CR100, a resistant pakchoi. Scale bars = 1 cm. The red arrow indicates the “clubroot” of SZQ.

**Figure 2 ijms-24-16779-f002:**
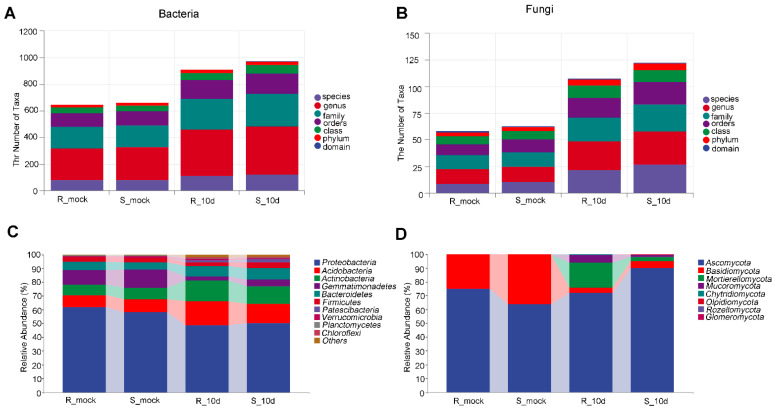
Microorganism composition analysis. (**A**,**B**) Numbers of taxa at different levels in bacteria (**A**) and fungi (**B**). (**C**,**D**) relative abundance of bacteria (**C**) and fungi (**D**) in different groups at phylum levels.

**Figure 3 ijms-24-16779-f003:**
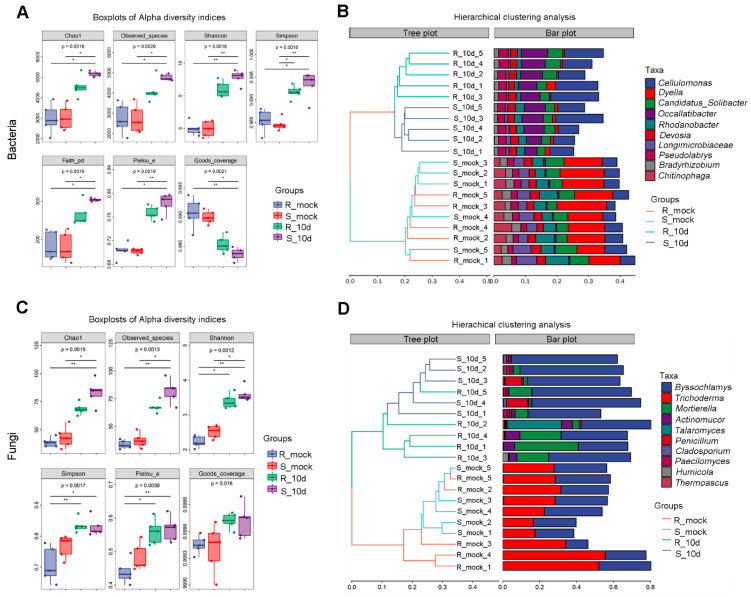
Microbial diversity analysis in bacteria and fungi. (**A**,**B**) Alpha diversity (**A**) and clustering analysis at genus levels (**B**) in bacteria. (**C**,**D**) Alpha diversity (**C**) and clustering analysis at genus levels of the top 10 taxa with the most abundance (**D**) in fungi. * *p* < 0.05, ** *p* < 0.01 (Dunn’s test).

**Figure 4 ijms-24-16779-f004:**
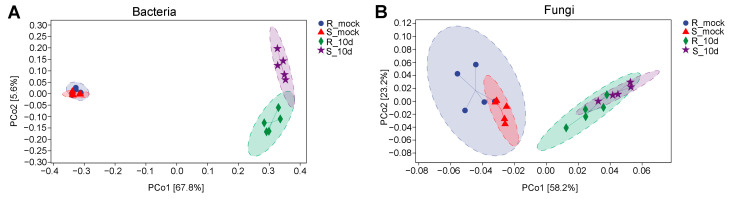
Principal coordinates analysis (PCoA) of bacteria (**A**) and fungi (**B**).

**Figure 5 ijms-24-16779-f005:**
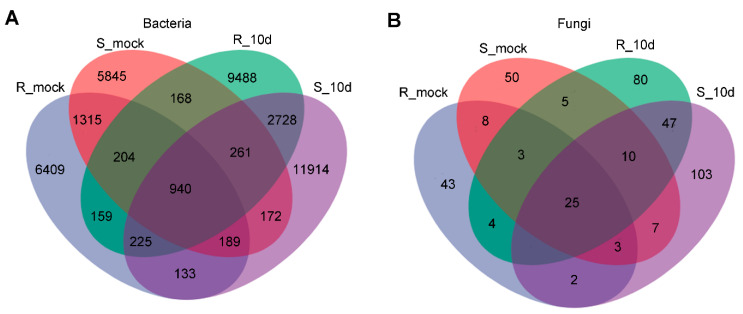
Venn analysis of bacterial (**A**) and fungal (**B**) ASVs among different groups.

**Figure 6 ijms-24-16779-f006:**
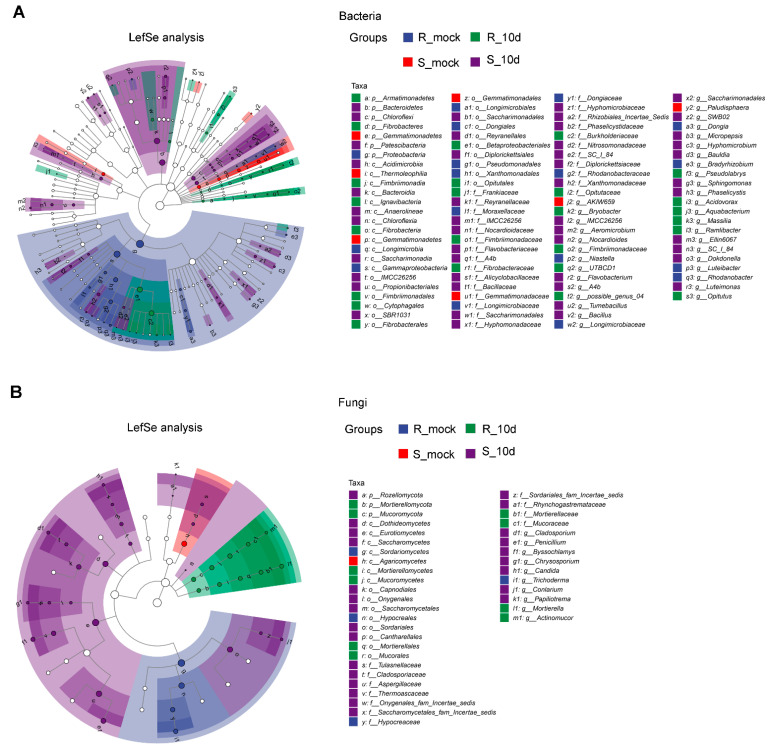
LefSe analysis between two cultivars with and without inoculation. (**A**) Bacterial LefSe analysis. LDA ≥ 2.82. (**B**) Fungal LefSe analysis. LDA ≥ 2. The diameter of each circle is proportional to the relative abundance of the taxon. The inner to outer circle corresponds to the level of the phylum to the genus. LefSe, LDA effect size; LDA, linear discriminant analysis. Hollow nodes represent taxa without significance. Nodes with colors represent taxa with significance among samples and high abundance in the corresponding group. p represents phylum; c represents class; o represents order; f represents family; g represents genus.

**Table 1 ijms-24-16779-t001:** The incidence of CR100 and SZQ.

Cultivar	Incidence	
	Uninoculated with *P. brassicae*	Inoculated with *P. brassicae*
CR100	0%	0%
SZQ	0%	100%

Note: Without *P. brassicae* inoculation, 50 seedlings for each cultivar; with *P. brassicae* inoculation, 33 SZQ and 52 CR100 seedlings were used for analysis.

## Data Availability

The bacterial diversity sequencing data presented in this study are openly available in NCBI under accession Nos. SRR 25888800-25888816 and SRR25888797-25888799. The associated BioProject is PRJNA1011212. Fungal diversity sequencing data are available in NCBI under accession No. SRR26051096-26051115. The associated BioProject is PRJNA1014988.

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
