# Peer review of "Changes in Diversity and Composition of Rhizosphere Bacterial and Fungal Community between Resistant and Susceptible Pakchoi under Plasmodiophora brassicae"

_ijms, 2023, doi:10.3390/ijms242316779_

Round 1
Reviewer 1 Report
Comments and Suggestions for Authors
The primary research question has been appropriately defined in the title and the objectives section of the paper. However, the discussion is not conducted at a sufficiently high level. I recommend expanding the discussion in my review. The problem and the tools employed are not groundbreaking in scientific terms, yet the economic significance of the issue necessitates research to minimize losses in the cultivation of Brassicaceae plants. Knowledge derived from these studies may contribute to the breeding of plants with increased resistance to the studied pathogen.
The paper attempts to depict variability using multidimensional statistical methods. If the journal had a more explicit statistical nature, such an analysis would not meet the requirements for interpreting the results. The International Journal of Molecular Sciences (IJMS) lacks a significant statistical inclination, so, for the majority of readers, the analysis may be considered moderately satisfactory.
The results of the experiment have been adequately described; however, there is room for improvement through enhancing the graphical presentation. Discrepancies in font sizes on Fig. 2 are noticeable, describing the axes, and there is a lack of proportionality and coherence between panels A-C and B-D. Additionally, grid lines are missing on Fig. 2B and 2D.
The discussion should be expanded to reference the latest scientific publications, enriching the analysis and strengthening the presented arguments.
Author Response
Dear Reviewer
We are very grateful to your kind and detailed comments which are very helpful to improve the manuscript. We have carefully revised the manuscript, which we hope meet with approval. Revised portions are marked in red in the manuscript. Revisions throughout the revised manuscript and the responds to comments have been made as followings:
Comment 1
The results of the experiment have been adequately described; however, there is room for improvement through enhancing the graphical presentation. Discrepancies in font sizes on Fig. 2 are noticeable, describing the axes, and there is a lack of proportionality and coherence between panels A-C and B-D. Additionally, grid lines are missing on Fig. 2B and 2D.
Response: Thank you for your kind advices. We have modified the font and image position on Fig.2. We contacted the Shanghai Personalbio Technology and they can only add grid line or trend line on Fig.2B and 2D. If necessary, we can delete trend line and add grid line on Fig.2B and 2D.
Comment 2
The discussion should be expanded to reference the latest scientific publications, enriching the analysis and strengthening the presented arguments.
Response: Thanks. We have revised the discussion mainly in line 224-235 and line 247-251.

Reviewer 2 Report
Comments and Suggestions for Authors
The manuscript entitled "Changes in diversity and composition of rhizosphere bacterial and fungal community between resistant and susceptible pakchoi's under Plasmodiophora brassicae" is interesting and has the potential to interest readers. The issues presented in this manuscript are consistent with the topics of the "International Journal of Molecular Science". The aims of the research were to analyze the diversity and community of rhizosphere bacteria and fungi of two different pakchoi cultivars to Plasmodiophora brassicae. The goals are original and well-defined, and the results obtained represent an advance in existing knowledge. The results are significant. A similar remark also applies to applications.
Detailed comments
1. The abstract needs to be fortified with numerical results.
2. The introduction part is very poor and lacks recent information; the majority of the introduction references must be included from 2016 to 2023 years. Highlight the novelty of your research here.
3. The "Material and methods" chapter requires several additions, as the reader cannot currently reproduce the experiment. In subsection "4.1 Pakchoi materials", please write how much soil was in each pot, what was its granulometric composition, and how long were the plants cultivated? Complete section "4.2 P. brassicae inoculation" with information: what mass of roots were washed with water and ground to extract resting spores. In section "4.3 Microbial diversity sequencing", specify what mass of soil was used to extract total microbial DNA.
4. Chapter "3. Discussion” requires additions. Now it is quite poor. Also, indicate guidelines for future research and limitations of this work.
5. Please, be sure that all the references cited in the manuscript are also included in the reference list and vice versa with matching spellings and dates.
Author Response
Dear Reviewer
We feel great thanks for your professional review work on your manuscript. These comments are valuable and helpful to improve the manuscript. According to your comments, we have revised our manuscript and changes were highlighted using red-colored text. Point-by-point responses are listed below:
Comment 1
The abstract needs to be fortified with numerical results.
Response: Thanks a lot. We revised the abstract in line 19-28 and line 25-28.
Comment 2
The introduction part is very poor and lacks recent information; the majority of the introduction references must be included from 2016 to 2023 years. Highlight the novelty of your research here.
Response: Thank you very much. We revised introduction in line 64-68, line 73-76, and line 81-88 and added several new references.
Comment 3
The "Material and methods" chapter requires several additions, as the reader cannot currently reproduce the experiment. In subsection "4.1 Pakchoi materials", please write how much soil was in each pot, what was its granulometric composition, and how long were the plants cultivated? Complete section "4.2 P. brassicae inoculation" with information: what mass of roots were washed with water and ground to extract resting spores. In section "4.3 Microbial diversity sequencing", specify what mass of soil was used to extract total microbial DNA.
Response: Thanks for your comments. We added the size of tray and each pot in 4.1 and purchasing company information. Excessive diseased roots were grounded and added with suitable water to prepare spore suspension for every experiment. The root mass was different each time, but the concentration of spore suspension was fixed and 1000 ml spore suspension was added into 1 kg soil before seeds were sowed into soil as mentioned in part 4.2. Therefore, it is difficult to say what mass of roots were washed with water and ground to extract resting spores. Regarding part 4.3, we added the method collecting rhizosphere soil and the mass of rhizosphere soil.
Comment 4
Chapter "3. Discussion” requires additions. Now it is quite poor. Also, indicate guidelines for future research and limitations of this work.
Response: Thanks a lot. We have made revision in discussion part mainly in line 224-235 and line 247-251.
Comment 5
Please, be sure that all the references cited in the manuscript are also included in the reference list and vice versa with matching spellings and dates.
Response: Thank you very much. We have checked all the references.
